# A Facile Synthesis of Noble-Metal-Free Catalyst Based on Nitrogen Doped Graphene Oxide for Oxygen Reduction Reaction

**DOI:** 10.3390/ma15030821

**Published:** 2022-01-21

**Authors:** Vladimir P. Vasiliev, Roman A. Manzhos, Valeriy K. Kochergin, Alexander G. Krivenko, Eugene N. Kabachkov, Alexander V. Kulikov, Yury M. Shulga, Gennady L. Gutsev

**Affiliations:** 1Institute of Problems of Chemical Physics of RAS, Acad. Semenov ave. 1, 142432 Chernogolovka, Russia; rmanzhos@yandex.ru (R.A.M.); kocherginvk@yandex.ru (V.K.K.); krivenko@icp.ac.ru (A.G.K.); en.kabachkov@gmail.com (E.N.K.); kulav@icp.ac.ru (A.V.K.); yshulga@gmail.com (Y.M.S.); 2Chernogolovka Scientific Center, Russian Academy of Sciences, 142432 Chernogolovka, Russia; 3Department of Physics, Florida A&M University, Tallahassee, FL 32307, USA

**Keywords:** oxygen reduction reaction, noble-metal-free catalysts, graphene oxide, melamine, ball-milling, N-doped

## Abstract

A simple method for the mechanochemical synthesis of an effective metal-free electrocatalyst for the oxygen reduction reaction was demonstrated. A nitrogen-doped carbon material was obtained by grinding a mixture of graphene oxide and melamine in a planetary ball mill. The resulting material was characterized by XPS, EPR, and Raman and IR spectroscopy. The nitrogen concentration on the N-bmGO surface was 5.5 at.%. The nitrogen-enriched graphene material (NbmGO has half-wave potential of −0.175/−0.09 V and was shown to possess high activity as an electrocatalyst for oxygen reduction reaction. The electrocatalytic activity of NbmGO can be associated with a high concentration of active sites for the adsorption of oxygen molecules on its surface. The high current retention (93% for 12 h) after continuous polarization demonstrates the excellent long-term stability of NbmGO.

## 1. Introduction

Fuel cells are considered as promising, renewable, and environmentally friendly energy sources. For the widespread use of fuel cells, however, it is necessary to solve several problems, one of which is related with fabrication of proper catalysts. Currently, in fuel cells with polymer electrolyte membranes, the active component of cathodic catalysts is presented by nanoparticles composed of platinum or its alloys which are deposited on carbon black [1,2,3]. Catalysts of this type have both advantages such as complete oxygen reduction and low overvoltage and obvious and fundamentally inevitable disadvantages such as sensitivity to impurities, high cost, limited resources, and low oxygen reduction reaction rates (ORR). It should be noted that the latter property restricts the load characteristics of fuel cells based on such catalysts [4,5,6]. One of the directions of research in the creation of ORR catalysts is the use of various types of carbon nanoforms (graphene-like structures, nanotubes, fullerenes, etc.) as carriers for Pt and its alloys. Another direction of research is based on the development of simple and effective methods for modifying the carbon structures themselves to create on their basis metal-free catalysts for the oxygen electroreduction in fuel cells.

Carbon nanostructures doped with *p*-elements are considered to be promising electrocatalysts for the oxygen reduction reaction [1,7,8]. The catalytic characteristics of modified graphene-like structures, for example, with nitrogen, are due to ability of their atoms to form a delocalized conjugated system with sp^2^-hybridized carbon, where a common positive charge on atoms of the carbon frame adjacent to nitrogen is created [9,10]. Note that in order to increase the productivity of catalysts based on graphene structures, it is necessary to increase the specific concentration of active ORR centers on the electrode surface accessible to electrolyte.

It was shown in many studies (see, for example, [8,11,12]), that the oxygen reduction reaction proceeds on such structures according to the four-electron mechanism, which was previously considered [13,14] to be characteristic only for platinum and platinum alloy catalysts. The production of nitrogen-doped carbon nanoform using standard methods is complicated by the possible toxicity of nitrogen precursors and their contamination of the final products as well as by the necessity to use expensive specialized equipment is required. Therefore, the development of non-standard methods for obtaining doped graphene-based catalysts presents an important task. In particular, a mechanochemical approach can be used since it allows doping of graphene structures with nitrogen atoms to be performed. In our case, it allows to create active ORR centers distributed over highly dispersed material. In other words, there is an increased electrode surface available for electrolyte. It should be noted that particles of such materials are less susceptible to agglomeration, in contrast to hydrophobic graphene sheets, which should contribute to a higher temporal stability of the electrocatalytic characteristics of such catalysts.

In the present work, we proposed a strategy for a simple one-step solid-state synthesis of nitrogen-enriched carbon powder (NbmGO) using inexpensive industrial precursors, namely graphene oxide (GO) and melamine. The synthesized material was characterized by methods of scanning electron microscopy, X-ray photoelectron spectroscopy, and infrared and Raman spectroscopies. In addition, NbmGO was tested as an ORR electrocatalyst and showed a higher efficiency (a decrease in ORR overvoltage and an increase in the contribution of complete oxygen reduction to water in the overall process) compared to the results obtained previously [15]. It should be noted that the number of studies on the synthesis of carbon materials and their modification using solid-phase methods is quite scarce, and restricted to the splitting of graphite [16,17] and the preparation of composites of carbon materials with transition metal oxides [18,19].

## 2. Experimental

### 2.1. Synthesis of Nitrogen-Doped Carbon Material

Graphene oxide was synthesized using a modified Hammers method [20] with chemical composition C_8_O_4.6_H_1.8_(H_2_O)_0.58_ and density ~1.2 g/cm^3^. Melamine C_3_N_6_H_6_ (99.9%, BASF SE, *Mannheim,* Germany) was used as a source of nitrogen (for details, see the Appendix A). The mechanochemical synthesis was carried out in a FRITSCH pulverisette-6 planetary mill with a grinding vessel and balls made of ZrO_2_. The internal diameter of the grinding vessel, the volume, and the ball diameter were 65 mm, 85 mL, and 5 mm, respectively. The GO/melamine ratio of reagents was 4:1, rotation speed was 400 rpm, and grinding time was 6 min. After grinding, the resulting powder was kept for 1 h in a 10% aqueous solution of ammonia, treated in an ultrasonic bath, and then it was centrifuged and washed with water 4–5 times to remove melamine residues.

### 2.2. Characterization

The sample images were acquired using a Zeiss LEO SUPRA-25 scanning electron microscope (Jena, Germany), and Raman spectra were recorded using a Bruker Senterra spectrometer Billerica, Billerica, MA, USA). The laser radiation wavelength was 532 nm, the radiation power at the measurement point was 1 mW, and the diameter of the analyzing laser beam was ∼1 μm. Infrared spectra of the powders were obtained using an FT-IR VERTEX 70v spectrometer (Billerica, MA, USA) in vacuum (50 scans with the resolution of 4 cm^−1^).

XPS spectra were obtained with the use of an electronic spectrometer for chemical analysis Specs PHOIBOS 150 MCD (Berlin, Germany). When recording the spectra, the vacuum in the spectrometer chamber did not exceed 2 × 10^−10^ Torr; the X-ray tube was equipped with a magnesium anode (Mg Kα radiation is 1253.6 eV) and the source power was 225 W. The survey spectrum was recorded in the range 0–1000 eV in the constant transmission energy mode (40 eV for the survey spectrum and 10 eV for individual lines). The survey spectrum was recorded with a step of 1 eV, while the spectra with individual lines with a step of 0.05 eV.

The ESR spectra of the powders were recorded at room temperature with a Bruker Elexsys II E 500 EPR spectrometer (Billerica, MA, USA) and an SE/X 2544 radio spectrometer (Radiopan, Poznan, Poland). The number of spins *N* and the *g*-factor were determined using the Xepr software package. To check the correctness of these procedures, a weighed quantity of CuSO_4_ × 5H_2_O and a DPPH sample with a *g*-factor of 2.0036 were used. The accuracy of concentration determination was ~15%. The electronic absorption spectra were obtained using a spectrophotometer PE-5400uf (Orenburg, Russia) and a Shimadzu UV-3101PC. The conductivity of the sample films was recorded on a potentiostat P-20X Elins (Orenburg, Russia) using a Micru XIDE1 thin-film Au-interdigitated electrode (90 pairs, 10/10 μm, electrode/gap).

### 2.3. Electrochemical Measurements

Voltammograms with linear potential sweep were measured in a three-electrode cell on a setup with the RRDE-3A rotating disk electrode s (ALS Co., Ltd., Naka-ku Sakai, Japan) using a potentiostat Autolab PGSTAT 302N (Metrohm Autolab, Utrecht, Holland) in an oxygen-saturated 0.1 M KOH solution with a potential sweep rate of *v* = 10 mV/s at electrode rotation frequencies ω = 360–6400 rpm. The current–voltage curves were analyzed using the Koutetsky–Levich equation [21]:(1)$$\frac{1}{j}=\frac{1}{j_{k}}+\frac{1}{j_{d}}$$
*j*_k_ = *n*F*kc*^0^(2)
*j*_d_ = 0.62*n*FD^2/3^ω^1/2^*υ*^−1/6^*c*^0^(3)
where *j*_k_ is the density of kinetic current, *j*_d_ is the density of limiting diffusion current, F is the Faraday number (F = 96,485 C/mol), *n* is the number of electrons participating in the electrode reaction, D is the coefficient of oxygen diffusion in a 0.1 M KOH solution (D = 1.9 × 10^−5^ cm^2^/s), *υ* is the kinematic viscosity of a 0.1 M KOH solution (*υ* = 0.01 cm^2^/s), and *c*^0^ is the volume concentration of dissolved oxygen (*c*^0^ = 1.2 mM in a 0.1 M KOH solution) [9,22].

A glassy carbon (GC) disk with a diameter of 3 mm, pressed into a PEEK polymer (ALS Co., Ltd., Naka-ku Sakai, Japan), served as a working electrode. The electrode surface was preliminarily polished with 1μm Al_2_O_3_ powder, then a drop of an aqueous suspension of bmGO or NbmGO with a volume of ~ 6 μL and a concentration of 1 mg/mL, containing 0.01 wt% Nafion, was applied and dried at room temperature. A platinum wire with an area of ~1 cm^2^ was used as an auxiliary electrode and an Ag/AgCl electrode filled with a saturated KCl solution was used as a reference electrode. All potentials (*E*) are given on the scale of the reference electrode. The bmGO deposited on the GC electrode was preliminarily electrochemically reduced during potential cycling (20–50 cycles) in an O_2_-saturated 0.1 M KOH solution in the *E* range from 50 mV to −1300 mV at a potential sweep rate of 50 mV/s. The catalyst stability was tested using chronoamperometry method at −250 mV for 12 h.

## 3. Results and Discussions

### 3.1. Structural Characterization of bmGO and NbmGO

#### 3.1.1. SEM

Figure 1 shows SEM images of samples of the starting graphene oxide (GO), ball milled graphene oxide (bmGO), and nitrogen-enriched carbon material (NbmGO) obtained by processing a mixture of GO and melamine in a ball mill. As can be seen from the figure, highly dispersed materials with a size of visible aggregates not exceeding 50 nm were formed by processing in the ball mill. This process increases the effective surface of the electrode accessible for electrolyte, and only small fragments of the original GO sheets can be observed in Figure 1b,c.

The elemental composition of carbon materials was calculated using analytical lines of the survey XPS spectrum (Appendix A). The oxygen concentration decreases markedly only for the NbmGO sample. The recovery of the sample is accompanied by an increase in the nitrogen content up to 5.5 at.% (Appendix A). The relative concentration of carbon in the NbmGO and bmGO samples increases insignificantly, by only 2.4−2.5 at.%. Although parent GO is an insulator, significant electronic conduction occurs in the films of the bmGO and NbmGO samples (Appendix A).

#### 3.1.2. XPS 

The shape of the C1*s* line in the spectra of the samples under study and the line fitting with four symmetrical Gaussian–Lorentz curves are presented in Figure 2a–c. The position of the main most intense peak (C1) is typical for *sp*^2^-carbon materials (Appendix A). The second most intense peak (C2) refers to carbon atoms that singly bonded with a nitrogen or oxygen atom of a hydroxyl (C–OH) or epoxy groups. The third peak (C3) can be attributed to the carbon atoms having two bonds with the oxygen atom (C=O or O–C–O). The last peak (C4) was assigned to the carbon atoms of the carboxyl groups (O–C=O) [23]. Note that the assignment of individual peaks was done in accordance with recommendations in [24].

According to the previous studies [8,25], nitrogen in a graphite-like matrix can be found in four or five configurations: pyridinic (N1, six-membered ring), pyrrolic (N2, five-membered ring), graphitic (N3/N4), and oxidized pyridinic (N5) (see Appendix A). The pyridinic and pyrrolic nitrogen atoms are located at the edges of a graphene sheet or at the defect sites. Nitrogen atoms of the N3 and N4 types replace carbon atoms in the graphite structure and differ by the location type. The nitrogen atom scan be located at the edges of a graphene sheet (N3) or in its center (N4). Nitrogen, which is a part of cyano and amino groups, may also be present [26].

The identification of surface nitrogen-containing groups can be carried out on the basis of an analysis of the fine structure of the N1*s* line in the high resolution XPS spectrum. According to the literature (see, e.g., [8,25] and references therein), pyridine nitrogen (N1) appears in the range 398.0–399.3 eV and pyrrole nitrogen (N2) appear in the range 399.8–401.2 eV) in the XPS spectra of nitrogen-doped carbon materials. It is worth noting that the N1*s* lines of amino (399.1 eV [25]) and cyano (399.3 eV [25]) groups are also located in this region; therefore, it is difficult to clearly identify it. The peak corresponding to the nitrogen atoms of the N4 type (inside a graphite sheet) is located at about 401 eV, and the peak corresponding to terminal graphite nitrogen (N3) is at 402.3 eV. Oxidized pyridine nitrogen (N5) corresponds to a peak at 402.8 eV while pyrrole nitrogen (N2) corresponds to a peak at 404.7 eV [25].

Three peaks can be distinguished in the N1*s* line of the NbmGO sample (see Figure 2d). When assigning these peaks, we came to a quite unexpected conclusion that the major contribution to the N1*s* line comes from the nitrogen atoms of the pyrrolic (N2) nitrogen. It is interesting to note that it was concluded in a model catalyst study that pyridinic nitrogen in graphite structures creates active centers for the oxygen reduction reaction [27].

#### 3.1.3. FTIR 

The FTIR spectra of melamine, initial graphene oxide, bmGO, and NbmGO are shown in Figure 3. It can be seen in the Figure that the IR spectrum of NbmGO differs from the IR spectra of the starting materials. Thus, absorption bands of stretching vibrations of N–H bonds, whose maxima in the spectrum of pure melamine are located at 3468, 3417, 3324, and 3121 cm^−1^, are lacking in the NbmGO spectrum. At the same time, the spectrum of NbmGO has absorption bands, which can be attributed to stretching vibrations of cyano groups in different environments (the region from 2350 to 1900 cm^−1^). Comparing the spectrum of NbmGO with the spectra of GO and bmGO, one can note that the absorption band due to the stretching C=O vibrations is practically absent in the NbmGO spectrum. Furthermore, one can notice a significant increase in the intensity of the absorption band associated with the vibrations of the C=C double bonds forbidden in the IR spectrum. According to the literature data, the band at 1360–1370 cm^−1^ can be attributed to vibrations of the C–OH bond, and the band at 1220 cm^−1^ corresponds to vibrations of C–O–C bonds. The band at 1060 cm^−1^ is attributed to the vibrations of alkoxy groups [28] (See Appendix A).

#### 3.1.4. Raman

The Raman spectra of the samples studied are shown in Figure 4. The spectra contain peaks designated as D, G, and 2D. It is well known that the 2D peak for single-layer graphene is a narrow peak, whose intensity exceeds the intensity of peak G. In the case of two-layer graphene, the intensity of the 2D peak decreases, while its half-width increases [29]. When the number of layers in graphene becomes more than five, the 2D peak disappears.

The presence of a 2D peak in the Raman spectra of our samples means that there are few-layer graphene-like structures. In addition, note the shift in the position of the G peak towards lower values in the GO→bmGO→NbmGO series (see Appendix A). A shift in the position of the G peak towards lower values during GO milling was also observed previously [30].

The ratio of the D and G band intensities (I_D_/I_G_) can be used to estimate the size of the sp^2^ domains of *L*_a_ in the basal plane [31]:*L*_a_ = (2.4 × 10^−10^) λ^4^ (*I*_D_/*I*_G_)^−1^(4)

Based on the ratio of the D/G peak intensities (see Appendix A), the size of *sp*^2^ domains is 19 nm (NbmGO), 20 nm (bmGO), and 22 nm (GO). Thus, milling leads to a slight increase in the number of defects and a decrease in the size of *sp*^2^ domains. The insertion of nitrogen atoms into the graphene oxide lattice also increases the number of defects and decreases the size of the graphene *sp*^2^ domains.

#### 3.1.5. ESR

The ESR spectra obtained for the samples under study are presented in Figure 5. As can be seen in the figure, the ESR spectra of all samples contains narrow singlet lines. The *g* factors of all samples are close to 2.0, which is typical for radicals where unpaired electrons of aromatic rings composed of carbon atoms occupy localized π-states [32,33].

The measured spin concentration (see Appendix A) for the NbmGO sample is 6.2 × 10^18^ PC/g, which is an order of magnitude higher than the concentration of the GO sample (5.4 × 10^17^ PC/g) and is approximately comparable to the spin concentration for the bmGO sample (5.5 × 10^18^ PC/g). This is an indication of a significantly larger number of defects on the surface of carbon structures of the samples processed in the planetary mill, which largely determines their electrocatalytic activity in ORR.

### 3.2. Electrochemical Analysis

The voltammogram dependencies obtained on the initial GC electrode and GC electrodes coated with bmGO and NbmGO in a 0.1 M KOH solution saturated with oxygen are shown in Figure 6a. To determine the number of electrons *n* participating in the ORR (see Figure 6b), the *j* and *E* dependences were measured at different speeds of electrode rotation. Figure 6c shows a series of such voltammogram curves for NbmGO and Figure 6d displays the dependence of *j* on ω plotted in the Koutetsky–Levich coordinates. From the slope of the *j*-ω dependences, the values of *n* were calculated at various values of *E* (see Figure 6b).

As can been seen from Figure 6a, the overpotential of the oxygen reduction reaction for bmGO and NbmGO significantly decreases compared to that of the initial GC. The half-wave potentials of the first oxygen reduction wave for GC, bmGO, and NbmGO are −365 mV, −225 mV, and −175 mV, respectively. In addition, an increase in the oxygen reduction current was observed when passing from the original GC electrode to the GC coated with bmGO and NbmGO.

On the voltammogram curve for bmGO (curve 2 in Figure 6a), two distinct waves which correspond to the predominant reduction of oxygen to hydrogen peroxide in the potential range from −250 to −500 mV (*n* ≈ 2.2–2.3) and water at E < −850 mV (*n* ≈ 4) can be distinguished. In the case of NbmGO (curve 3 in Figure 6a), the first wave includes a segment of linear growth of the cathodic current of in the potential range from −400 to −750 mV, while the second wave is characterized by reaching the limit at E < −900 mV and corresponds to the limiting diffusion current of complete oxygen reduction to water. The corresponding value of the limiting diffusion current density calculated by using Equation (3) is *j*_d_ = −6.4 m /cm^2^. It should be noted that there is also a linear increase in the number of electrons participating in the oxygen reduction reaction from 2.8 to 4.0 (Figure 6b) in the potential range from −250 to −850 mV.

In the case of NbmGO O_2_ is reduced to H_2_O (*n* ≈ 2.8) even at low overpotentials (*E* = −200 mV) in addition to its reduction to hydrogen peroxide with a gradual increase in the contribution of this process until the complete reduction of oxygen to water at *E* < −850 mV (*n* ≈ 4). In general, bmGO and NbmGO are characterized by a significant decrease in the ORR overvoltage and higher *n* values as compared to glassy carbon. This is an indicator of a high concentration of active centers for the adsorption of both oxygen molecules on the surface of bmGO and NbmGO and intermediates of its reduction, which can be surface defects and edge regions of graphene-like structures [34], quinone groups [35], and pyridine nitrogen atoms in the case of the NbmGO sample [27,36], as shown earlier [37].

In addition to high catalytic performance, the ideal electrode material should have excellent long-term stability, which can be evaluated by prolonged chronoamperometry. A chronoamperometry test for the NbmGO catalyst was carried out at 2000 rpm in O_2_-saturated 0.1 M KOH solution (see Appendix A). The high current retention of 93 % after continuous polarization at −250 mV during 720 min clearly demonstrates the excellent stability of NbmGO.

## 4. Conclusions

The present work reports on a simple solid-phase method for the synthesis of an effective metal-free electrocatalyst for the oxygen reduction reaction from inexpensive industrial materials, namely, graphene oxide, and melamine. Our method allows the doping of graphene structures with nitrogen atoms and obtaining highly dispersed materials with increased effective electrode surfaces accessible by electrolytes. For the material obtained, a decrease in the ORR overvoltage and an increase in the contribution of the complete reduction of oxygen to water compared to those for glassy carbon are shown. The observed catalytic activity is determined by the high concentration of active sites for adsorption of oxygen molecules and intermediate intermediates of its reduction. The NbmGO long-term stability can be attributed to a small loss of active sites during the test.

## Figures and Tables

**Figure 1 materials-15-00821-f001:**
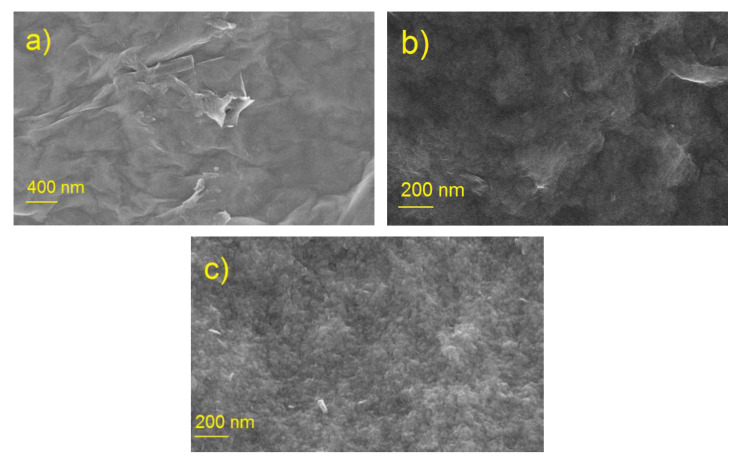
SEM images of GO (**a**), bmGO (**b**), and NbmGO (**c**).

**Figure 2 materials-15-00821-f002:**
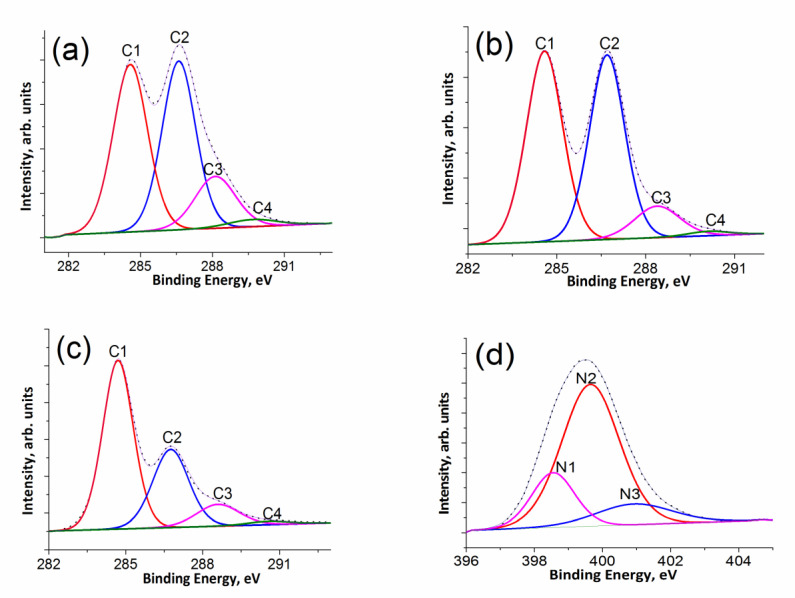
C1*s* lines in the XPS spectra of GO (**a**), bmGO (**b**), NbmGO (**c**), and N 1*s* lines of NbmGO (**d**).

**Figure 3 materials-15-00821-f003:**
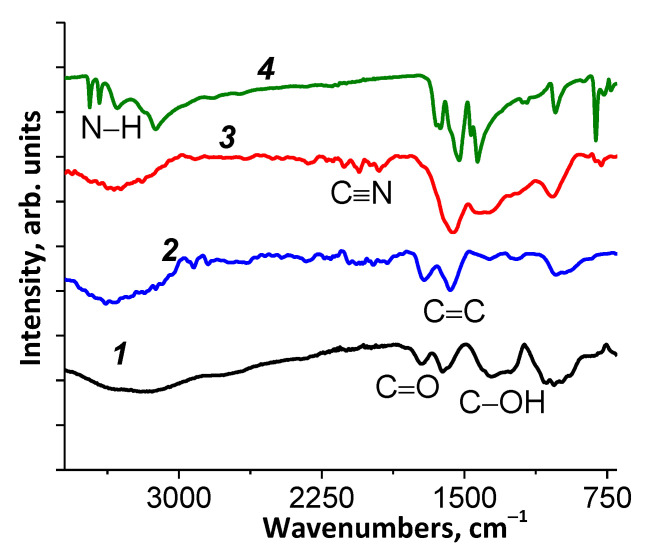
FTIR spectra of GO (***1***), bmGO (***2***), NbmGO (***3***), and melamine (***4***).

**Figure 4 materials-15-00821-f004:**
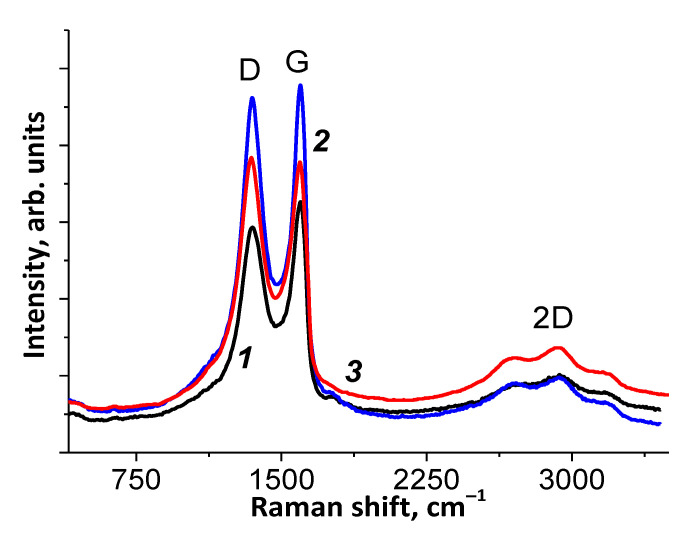
Raman spectra of GO (***1***), bmGO (***2***), and NbmGO (***3***). See the text for the peak designation.

**Figure 5 materials-15-00821-f005:**
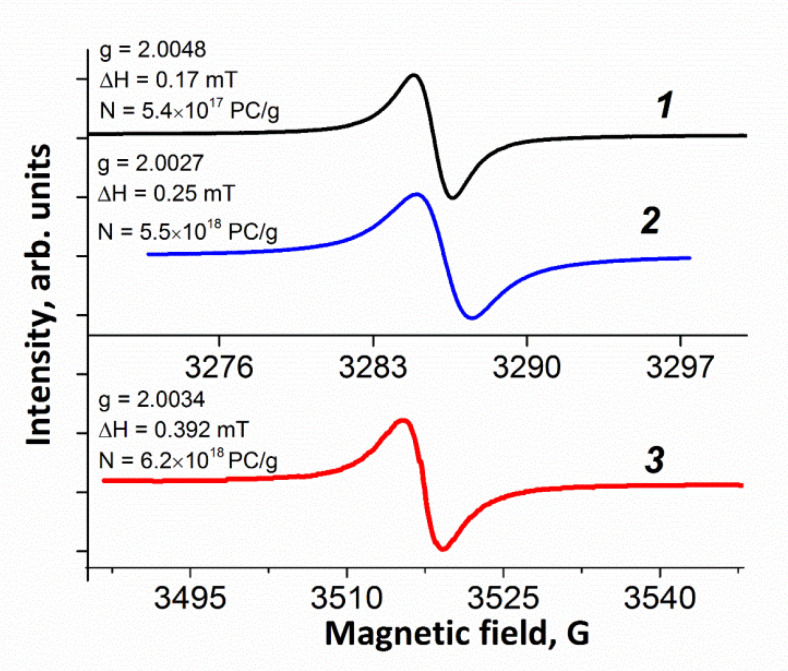
ESR spectra of GO (***1***), bmGO (***2***), and NbmGO **(*3***) obtained at room temperature.

**Figure 6 materials-15-00821-f006:**
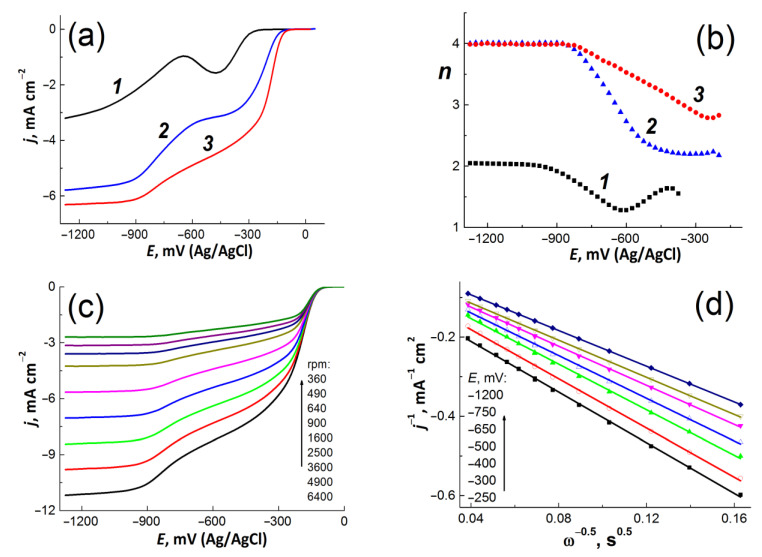
(**a**) Linear sweep voltammograms of O_2_-saturated 0.1 M KOH solution for the bare GC electrode (curve 1) and GC coated with bmGO (curve 2) and NbmGO (curve 3), *v* = 10 mV/s, ω = 2000 rpm. (**b**) Dependencies of the electron transfer number *n* on the potential E for the bare GC electrode (curve 1), GC covered with bmGO (curve 2), and NbmGO (curve 3). (**c**) Voltammograms for NbmGO measured at different speed of electrode rotations. (**d**) Corresponding *j-*ω dependences in the Koutecký–Levich coordinates.

## Data Availability

Data available in a publicly accessible repository.

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
