# Peer review of "A Facile Synthesis of Noble-Metal-Free Catalyst Based on Nitrogen Doped Graphene Oxide for Oxygen Reduction Reaction"

_materials, 2022, doi:10.3390/ma15030821_

Round 1

Reviewer 1 Report

In this work, Vasiliev and co-workers report a simple solid-phase approach for the synthesis of an effective metal-free electrocatalyst for the oxygen reduction reaction by using graphene oxide and melamine. This work is interesting and might be useful for research in fuel cells and other energy-related fields. However, this work has several major concerns, which need to be considerably improved. So I recommend this work after major revision.

  1. The ABSTRACT part is missing.
  2. The introduction is not well written, and several essential parts are missing. For instance, the research background on the electrocatalyst of solid carbon materials for oxygen reduction, the main bottleneck of the current research, etc.
  3. In the XPS analysis, the peaks with the same binding energy should be noted as the same color as they have the same physics meanings. Currently, the figure looks quite confusing.
  4. The method that the author mentioned in the Raman part by using the ratio of the D and G band intensities to estimate the size the sp2 domains is not precise. In the evolution of the ID/IG, actually there are two stages, one is the transformation from graphene to polycrystalline graphene and one is from polycrystalline graphene to amorphous carbon, and each stage has a different equation to estimate the sp2 domains. So this part needs to be made clear.
  5. The catalysis efficiency needs to be compared with other reported research.

Author Response

Reviewer #1:

  1. The ABSTRACT part is missing.

Authors reply:

The MS abstract is updated.

Reviewer #1:

  1. The introduction is not well written, and several essential parts are missing. For instance, the research background on the electrocatalyst of solid carbon materials for oxygen reduction, the main bottleneck of the current research, etc.
  2. Authors reply:

In the new version of the manuscript, the introduction has been rewritten with taking into account this comment.

Reviewer #1:

  1. In the XPS analysis, the peaks with the same binding energy should be noted as the same color as they have the same physics meanings. Currently, the figure looks quite confusing.

Authors reply:

In the new version of the manuscript, the peaks with the same binding energies have the same color.

 Reviewer #1:

  1. The method that the author mentioned in the Raman part by using the ratio of the D and G band intensities to estimate the size the sp2 domains is not precise. In the evolution of the ID/IG, actually there are two stages, one is the transformation from graphene to polycrystalline graphene and one is from polycrystalline graphene to amorphous carbon, and each stage has a different equation to estimate the sp2 domains. So this part needs to be made clear.

Authors reply:

In our work, a nitrogen-doped carbon material was obtained by grinding a mixture of graphene oxide with melamine in a planetary ball mill, and partial amorphization, reduction, and doping of graphene oxide occurs during the grinding process. Thus, the whole process is difficult to separate into the stages indicated by the Reviewer. However, it was  known that for the initial graphene oxide the La value was in the range from 10 nm to 30 nm and therefore the formula ID/IG ∝ 1/La can be used for the initial assessment. Our calculations whose results are presented in Table S3 of the Supporting Information led us to the conclusion that the La value does not diminish sufficiently to use the relation of Ferrari and Robertson ID/IG ∝ (La)2.

Reviewer #1:

  1. The catalysis efficiency needs to be compared with other reported research.

Authors reply:

The catalytic activity of the material obtained in the present work is compared with that of obtained in other studies (see Table S6 of the Supporting Information). Our catalyst NbmGO exhibits similar or even better activity in terms of the half-wave potential (Table SI6), but slightly loses in terms of the number of electrons involved in ORR (in comparison with similar catalysts reported in the literature).

We thank the Reviewer for the helpful comments.

Reviewer 2 Report

In the manuscript entitled "A Facile Synthesis of Noble-Metal-free Oxygen Reduction Reaction Catalyst Based on Nitrogen Doped Graphene Oxide", the author reported that the preparation of ‘nitrogen-enriched carbon powder (NbmGO) using inexpensive industrial precursors, namely, graphene oxide (GO) and melamine’ and used as an ORR catalyst. The results seem interesting. However, some minor revisions and more expressions are still needed to be published in this Journal. Here are some comments. 

  1. In the introduction part, it will be good if you can improve the quality of scientific way of writing, for example in the second paragraph of the introduction, ‘Catalysts of this type have both advantages such as complete oxygen reduction and low overvoltage and obvious and fundamentally inevitable disadvantages such as sensitivity to impurities, high cost, limited resources, and low oxygen reduction reaction rates (ORR)’. The sentence is not very clear.
  2. It will be good if you can cite the most relevant article regarding N-doped graphene, for example, Catalyst-Free Synthesis of Nitrogen-Doped Graphene via Thermal Annealing Graphite Oxide with Melamine and Its Excellent Electrocatalysis by Sheng et al.
  3. It will be more impressive if you can provide a high-resolution transmission electron micrograph (HRTEM) and PXRD pattern of GO before and after N doping.
  4. As you have mentioned in reference 14 (previous article from your group), what significant difference does it make in ORR activity, and what is the exact scientific reason behind this? The reason should be addressed.
  5. How is the stability of catalyst? Recommending for Chrono-amperometry (CA) measurements.

Author Response

Reviewer #2:

  1. In the introduction part, it will be good if you can improve the quality of scientific way of writing, for example in the second paragraph of the introduction, ‘Catalysts of this type have both advantages such as complete oxygen reduction and low overvoltage and obvious and fundamentally inevitable disadvantages such as sensitivity to impurities, high cost, limited resources, and low oxygen reduction reaction rates (ORR)’. The sentence is not very clear.

Authors reply:

Тhe introduction has been rewritten with taking into account these comments of the Reviewer.

 Reviewer #2:

  1. It will be good if you can cite the most relevant article regarding [Sheng, Z-H.; Shao, L.; Chen, J-J.; Bao, W-J.; Wang, F-B.; Xia, X-H. Catalyst-Free Synthesis of Nitrogen-Doped Graphene via Thermal Annealing Graphite Oxide with Melamine and Its Excellent Electrocatalysis, ACS Nano, 2011, vol. 5, p. 4350–4358].

Authors reply:

The paper indicated by the Reviewer are all cited in the new version of the manuscript.

Reviewer #2:
3. It will be more impressive if you can provide a high-resolution transmission electron micrograph (HRTEM) and PXRD pattern of GO before and after N doping.

Authors reply:

Unfortunately, we are unable to perform HRTEM and PXRD measurements at the present time. We believe that the main results obtained are sufficiently confirmed by the measurements described in the current MS version.

Reviewer #2:
4. As you have mentioned in reference 14 (previous article from your group), what significant difference does it make in ORR activity, and what is the exact scientific reason behind this? The reason should be addressed.

Authors reply:

Our previous article was a preliminary short communication. The current work reports the results obtained for new samples, and the measurements were performed by using additional experimental methods. In addition, the present study addresses the effects on catalytic activity of carbon material (reduction of ORR overvoltage) due to both defects and doped nitrogen. To increase the productivity of catalysts based on graphene structures, it is necessary to increase the specific concentration of active ORR centers on the electrode surface accessible to the electrolyte. The following statements are added into the introduction:

“In particular, a mechanochemical approach can be used since it allows doping of graphene structures with nitrogen atoms to be performed. In our case, it allows to create active ORR centers distributed over highly dispersed material. In other words, there is an increased electrode surface available for electrolyte. It should be noted that particles of such materials are less susceptible to agglomeration, in contrast to hydrophobic graphene sheets, which should contribute to a higher temporal stability of the electrocatalytic characteristics of such catalysts. “

Reviewer #2:
5. How is the stability of catalyst? Recommending for Chrono-amperometry (CA) measurements.

Authors reply:

The catalyst stability is addressed to in the new version of the manuscript as following.

In addition to high catalytic performance, the ideal electrode material should have excellent long-term stability, which can be evaluated by prolonged chronoamperometry. A chronoamperometry test for the NbmGO catalyst was carried out at 2000 rpm in O2-saturated 0.1 M KOH solution (see Fig. S5 in the Supporting Information). The high current retention of 93 % after continuous polarization at –250 mV during 720 min clearly demonstrates the exceptional stability of NbmGO.

We thank the Reviewer for the helpful comments.

Reviewer 3 Report

Reviewers' comments:

In the manuscript, the authors reported a simple one-step solid-state synthesis of nitrogen-enriched carbon powder (NbmGO) using graphene oxide (GO) and melamine. Generally, current work is well carried out but the authors should try to emphasize better the importance of current manuscript in order to attract the readership of this journal. Besides, the research work in this manuscript can be published in this journal after the authors consider the following major points.

.

TECHNICAL PART

  1. The English language needs significant attention and substantial improvement as several sentences are not quite clear.

TITLE MANUSCRIPT

  1. Please change and revise the title of manuscript. The title of the manuscript can suggested as “A Facile Synthesis of Noble-Metal-free Catalyst Based on Nitrogen Doped Graphene Oxide for Oxygen Reduction Reaction”.

ABSTRACT PART

  1. The authors should highlight Hypothesis, Experiments and Findings. The preferred format for the Abstract should be used in order to attract the readership of this journal.
  2. The Abstract also can be accompanied by numerical data/quantitative information to support the statements.
  3. Please revise the whole abstract.

INTRODUCTION PART

  1. The introduction part is too casual, not well written and don’t really have the flow of explaination. Please revise the whole introduction part. In addition, some of the sentences are very confuse and not really scientific discussion.
  2. How this reported research work is different from other published work?
  3. Please highlight the novelty of this study properly. The novelty based on the approach or product or both?

EXPERIMENTAL PART

  1. Every chemicals/reagents used in this study should be listed in details (Provide the purity, brand, manufacturer country of all reagents.).
  2. Please make revision in term of English language for all the procedure.

RESULTS AND DISCUSSION PART

  1. In general, the discussion of the obtained results for the samples are very poor and not really scientific. The authors should elaborate the discussion in detail for every analysis results.
  2. Please double check your FTIR spectra and need to discuss in detail.
  3. Please list of the transmission bands (cm-1) and functional groups obtained from FTIR for every samples in the table form. Please discuss in depth regarding the FTIR result.
  4. Under XPS, Raman and FTIR discussion, please give and cite the references to support the statement mentioned above. The suggested references can be cited such as Functional Materials Letters 8 (2) (2015): 1550026, Materialia 6 (2019): 100344, Physica E: Low-dimensional Systems and Nanostructures 131 (2021): 114727, Materials Chemistry and Physics 278 (2022): 125629.
  5. Under ESR discussion, the authors stated “As can be seen in the figure, the ESR spectra of all samples contains a narrow line.” Narrow line or narrow peak?
  6. Please revise Figure 6, all the graph should be in a figure, don’t be separated between a, b and c, d.
  7. The discussion of FESEM results for every samples are not sufficient. Please revise.
  8. There have repeated words (see line 268). Please check and revise.
  9. Please provide the EDX results for the samples.
  10. What is the new finding in this research which is completely different from other published articles on the Electrocatalyst towards ORR?
  11. The contribution with respect to other similar works appeared in literature (compare results in a table) and the possible future commercial application of the as-prepared nanocomposite electrocatalyst towards the fuel cell performance should be point out.

CONCLUSION PART

  1. Manuscripts published in this journal must explain the significant advances provided in approaches and understanding compared to previous literature, and/or demonstrate convincingly potential in new applications. The conclusions of your paper are especially important for this. Therefore, please try to sharpen this further. The optimal conclusion should include: A summary of your findings, A synopsis of your new concepts and innovations, A brief restatement of your hypotheses, Your vision for future work.
  2. There have repeated words. Please check and revise.

Author Response

TECHNICAL PART

Reviewer #3:

The English language needs significant attention and substantial improvement as several sentences are not quite clear.

 Authors reply: We tried our best to comply with this remark.

TITLE MANUSCRIPT

Reviewer #3:

Please change and revise the title of manuscript. The title of the manuscript can suggested as “A Facile Synthesis of Noble-Metal-free Catalyst Based on Nitrogen Doped Graphene Oxide for Oxygen Reduction Reaction”.

Authors reply:  Thanks for advising a better MS title. The recommended change is done.

ABSTRACT PART

Reviewer #3:

  1. The authors should highlight Hypothesis, Experiments and Findings. The preferred format for the Abstract should be used in order to attract the readership of this journal.
  2. The Abstract also can be accompanied by numerical data/quantitative information to support the statements
  3. Please revise the whole abstract.

Authors reply:

We revised the abstract with taking into account these remarks. In particular, numerical values characterizing the properties of the catalyst were added.

INTRODUCTION PART

Reviewer #3:

  1. The introduction part is too casual, not well written and don’t really have the flow of explaination. Please revise the whole introduction part. In addition, some of the sentences are very confuse and not really scientific discussion.
  2. How this reported research work is different from other published work?
  3. Please highlight the novelty of this study properly. The novelty based on the approach or product or both?

Authors reply:

The introduction has been rewritten in accordance with the suggestions of this and other Reviewers. It was underlined that the novelty is the new approach developed by the MS authors.

EXPERIMENTAL PART

Reviewer #3:

  1. Every chemicals/reagents used in this study should be listed in details (Provide the purity, brand, manufacturer country of all reagents.).
  2. Please make revision in term of English language for all the procedure.

 Auhors reply:

The  GO was synthesized using a well-known technique and the details of our synthesis are presented in the ESI. The melamine purity is reported in the MS text. The English editing was performed.

RESULTS AND DISCUSSION PART

Reviewer #3:

  1. In general, the discussion of the obtained results for the samples are very poor and not really scientific. The authors should elaborate the discussion in detail for every analysis results.
  2. Please double check your FTIR spectra and need to discuss in detail.
  3. Please list of the transmission bands (cm-1) and functional groups obtained from FTIR for every samples in the table form. Please discuss in depth regarding the FTIR result.
  4. Under XPS, Raman and FTIR discussion, please give and cite the references to support the statement mentioned above. The suggested references can be cited such as Functional Materials Letters 8 (2) (2015): 1550026, Materialia, 6 (2019): 100344, Physica E: Low-dimensional Systems and Nanostructures, 131 (2021): 114727, Materials Chemistry and Physics, 278 (2022): 125629.
  5. Under ESR discussion, the authors stated “As can be seen in the figure, the ESR spectra of all samples contains a narrow line.” Narrow line or narrow peak?
  6. Please revise Figure 6, all the graph should be in a figure, don’t be separated between a, b and c, d.
  7. The discussion of FESEM results for every samples are not sufficient. Please revise.
  8. There have repeated words (see line 268). Please check and revise.
  9. Please provide the EDX results for the samples.
  10. What is the new finding in this research which is completely different from other published articles on the Electrocatalyst towards ORR?
  11. The contribution with respect to other similar works appeared in literature (compare results in a table) and the possible future commercial application of the as-prepared nanocomposite electrocatalyst towards the fuel cell performance should be point out.

 Authors reply:

  1. The results and discussion part have been rewritten.
  2. The FTIR spectra discussion have been revised in the new version of the manuscript.
  3. All transmission bands and functional groups obtained uing FTIR are presented in the table S6.
  4. We cited two of the recommended papers [Hanifah, F.R.; Jaafar, J.; Aziz, M.; Ismail, A.F.; Othman, M.H.D.; Rahman, M.A.; Norddin, M.N.A.M.; Yusof, N.; Salleh, W.N.W. Efficient reduction of graphene oxide nanosheets using Na2C2O4 as a reducing agent, Functional Materials Letters, 2015, 8, 155026, 5p. and Hanifah, M.F.R.; Jaafar, J.; Othman, M.H.D.; Ismail, A.F.; Rahman, M.A.; Yusof, N.; Salleh, W.N.W.; Aziz, F. Facile synthesis of highly favorable graphene oxide: Effect of oxidation degree on the structural, morphological, thermal and electrochemical properties, Materialia, 2019, 6, 100344, 12p]. In two other articles recommended (Physica E: Low-dimensional Systems and Nanostructures, 131 (2021): 114727, and Materials Chemistry and Physics, 278 (2022): 125629) all XPS spectra contain an intense peak designated as C–C (sp3) at binding energy 284.5 eV which corresponds to a diamond-like peak. The presence of diamond-like peaks contradicts the generally accepted concepts on the structure of GO and rGO.
  5. The rewritten sentence reads “As can be seen in the figure, the ESR spectra of all samples are narrow singlet lines.”
  6. Figure 6 is revised.
  7. The discussion of the FESEM results has been rewritten in the new version of the manuscript.
  8. Thanks for the comment. Repeated words in the new version of the manuscript have been removed.
  9. The accuracy of determination of the concentration of light elements by the EDX method is very low; therefore, we did not provide these data.
  10. This work reports a new approach to synthesizing carbon materials suitable for catalyzing ORR. The present study explores effects of both defects and nitrogen on the catalytic activity of carbon materials (reduction of the ORR overvoltage). To increase the productivity of catalysts based on graphene structures, it is necessary to increase the specific concentration of active ORR centers on the electrode surface accessible to the electrolyte. The following statements are added into the introduction: “In particular, a mechanochemical approach can be used since it allows doping of graphene structures with nitrogen atoms to be performed. In our case, it allows to create active ORR centers distributed over highly dispersed material. In other words, there is an increased electrode surface available for electrolyte. It should be noted that particles of such materials are less susceptible to agglomeration, in contrast to hydrophobic graphene sheets, which should contribute to a higher temporal stability of the electrocatalytic characteristics of such catalysts. “
  11. The catalytic activity is compared with that from other studies in Table S6 of the ESI. Our catalyst NbmGO exhibits similar or even better activity in terms of the half-wave potential (Table S6), but slightly loses in terms of the number of electrons involved in ORR (in comparison with similar catalysts reported in the literature).

          The possibility of commercial application is not yet evident, and further fundamental research is required.

CONCLUSION PART

Reviewer #3:

  1. Manuscripts published in this journal must explain the significant advances provided in approaches and understanding compared to previous literature, and/or demonstrate convincingly potential in new applications. The conclusions of your paper are especially important for this. Therefore, please try to sharpen this further. The optimal conclusion should include: A summary of your findings, A synopsis of your new concepts and innovations, A brief restatement of your hypotheses, Your vision for future work.
  2. There have repeated words. Please check and revise.

 Authors reply:

The conclusion has been revised.

We have been deleted repeated words.

Thanks to the Reviewer for helpful comments!

Round 2

Reviewer 1 Report

The paper looks quite good now and I suggest publishing it.